# Unsupervised Image Enhancement Method Based on Attention Map Network Guidance and Attention Mechanism

**Mengfei Wu** [1,2]**, Taiji Lan** [1]**, Xucheng Xue** [1,*] **and Xinwei Xu** [1,2]

1   Changchun Institute of Optics, Fine Mechanics and Physics, Chinese Academy of Sciences, Changchun 130033, China
2   School of Optoelectronics, University of Chinese Academy of Sciences, Beijing 100049, China
*   Correspondence: xuexucheng@ciomp.ac.cn

**Abstract:** Low-light image enhancement is a crucial preprocessing task in complex vision tasks. It directly impacts object detection, image segmentation, and image recognition outcomes. In recent years, with the continuous development of deep learning techniques, an increasing number of image enhancement methods based on deep learning have emerged. However, due to the high cost of data collection and the limited content of supervised learning datasets, more and more scholars have shifted their focus to the field of unsupervised image enhancement. Unsupervised image enhancement methods do not require paired images of the same scene during the training process, which greatly reduces the threshold for network training. Nevertheless, current unsupervised methods still suffer from issues such as unstable enhancement effects and limited generalization ability. To address these problems, we propose an improved low-light image enhancement method. The proposed method employs the LSGAN as the training architecture and utilizes an attention map network to dynamically generate attention maps that best fit the network enhancement task, which can effectively improve the generalization ability and enhancement performance of the network. Additionally, we adopt an attention mechanism to enhance the subtle details of the image features. Regarding the network training, considering that the traditional convolutional neural network discriminator may not provide effective guidance to the generator in the early stages of training, we propose an improved discriminator structure. The experimental results demonstrate that our method can achieve good enhancement performance on different datasets and has practical value. Although our method has advantages in enhancing low-light images, it also has certain limitations, such as the network size not meeting the requirements for lightweight models and the potential for further improvement under extremely low-light conditions. We will strive to address these issues as comprehensively as possible in our future research.

**Keywords:** image enhancement; neural network; attention mechanism; attention map network; LSGAN

## 1. Introduction

As one of the primary sources of information available to humans, images occupy a critical position in the entire information society. With the rapid development of hardware technology in recent years, the quality of images generated by imaging devices has become increasingly high. These high-quality images not only provide people with better visual experiences but also greatly promote the development of other vision tasks, such as object detection, recognition, and image segmentation. However, images captured in low-light environments still face many problems, including severe contrast and brightness deficiencies and significant loss of detail information. Therefore, it is crucial to use appropriate algorithms to enhance the contrast, brightness, and details of low-light images.

Early researchers conducted research based on the characteristics of low-light images and proposed many image enhancement algorithms, such as the histogram equalization and Retinex-based enhancement algorithms, which are still widely used in the industry.

In recent years, with the development of deep learning technology, more and more people have begun to consider introducing deep learning technology into the field of image enhancement. Unlike traditional methods, deep learning-based image enhancement is a data-driven image enhancement technique and therefore has many advantages. Currently, deep learning-based image enhancement methods can be roughly divided into three categories based on their principles: supervised learning, unsupervised learning, and self-supervised learning. However, the majority of image enhancement algorithms rely solely on supervised training using paired datasets, which inherently suffers from limitations. This approach requires paired images in the same location with different lighting conditions, which undoubtedly increases the cost of data acquisition. Moreover, paired images of fixed scenes are unlikely to include moving objects or humans, resulting in a lack of image content diversity. Consequently, training with such data may lead to poor generalization of the network. Therefore, some scholars have proposed unsupervised image enhancement methods to reduce the data acquisition threshold. However, some unsupervised methods still have issues, such as generating attention maps using fixed patterns or using unreasonable attention networks in the network architecture, which negatively impact the quality of enhanced images. In this study, we address the aforementioned issues and propose our own solution by improving the relevant methods. Figure 1 presents a comparison of the enhancement effects achieved by the methodology proposed in this article and those obtained by traditional methods.

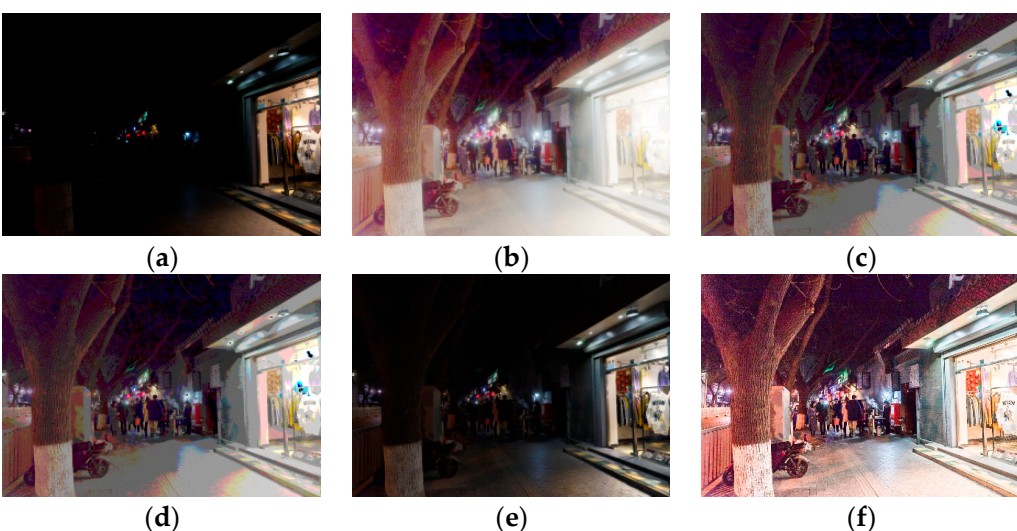

**Figure 1.** Enhancement result of traditional methods: (**a**) original, (**b**) MSR, (**c**) log transform, (**d**) CLAHE, (**e**) gamma, and (**f**) ours.

## 2. Related Works

### 2.1. Traditional Image Enhancement Methods

Image enhancement is an important image processing technique in the field of computer vision, with the primary objective of making the enhanced image more suitable for human visual perception or computer processing. Prior to the widespread application of deep learning techniques, most image enhancement tasks were solved using traditional methods. Traditional methods have advantages such as strong algorithm robustness, good real-time performance, and low computational resource requirements. Currently, the most commonly used image enhancement methods are largely based on histogram equalization and the Retinex theory.

Histogram equalization (HE) is an image processing technique that improves image contrast by stretching the grayscale levels. The distribution of grayscale levels in an image processed by HE is more uniform, which is an approximation of the maximum entropy theory. However, this approach has certain limitations, as it may cause over-exposure of the image and damage its fine details. To address this issue, scholars have proposed

various improved methods. For instance, Pizer et al. introduced the concept of adaptive histogram equalization (AHE) [1] using a moving window, but AHE may cause severe block effects. Then, they proposed contrast-limited adaptive histogram equalization (CLAHE) [2] based on AHE, which clips the histogram's grayscale levels to achieve limited contrast enhancement. Subsequently, Debdoot Sheet et al. [3] proposed a modified HE method called BPDHE, which fills the missing values in the histogram using linear interpolation and filters the histogram with a Gaussian filter. Kansal et al. [4] suggested controlling the shape of the cumulative distribution function (CDF) in HE to balance brightness and contrast. Chang et al. [5] incorporated the dual gamma correction method into CLAHE to achieve more natural and effective enhancement. These methods limit the over-enhancement of HE while improving the image's fine details and visual effects.

The Retinex theory has been widely used in the field of image enhancement and is currently one of the most commonly used techniques. The Retinex theory was first proposed by Edwin H. Land [6,7] in 1963, and the key to such methods is to estimate the illumination component of the original image and then calculate the reflective properties of objects. Subsequently, Jobson et al. proposed the Single-Scale Retinex (SSR) [8] method based on Gaussian center surround functions. While the SSR algorithm has good enhancement effects for color images, it is difficult to balance dynamic range compression and color consistency. To address this issue, the Multi-Scale Retinex (MSR) was proposed in Ref. [9]. MSR is essentially a weighted average of multiple different scale SSR. Following MSR, Rahman et al. further proposed the Multi-Scale Retinex with Color Restoration (MSRCR) [10] theory based on color restoration factors. The advantage of the MSRCR algorithm is that it can analyze and process the image according to its feature scale. Compared with MSR and SSR, MSRCR can enhance the color vividness of the image and suppress the nonlinear distortion to some extent. However, MSRCR is sensitive to the lighting conditions of the input image and usually has poor processing effects on overexposed or underexposed images. In recent years, other scholars have proposed new algorithms based on the Retinex theory. Ng et al. [11] proposed a total variation (TV) model for Retinex theory, which utilizes TV regularization to preserve image smoothness and details, and employs the Retinex principle to enhance image contrast and brightness. Sun et al. [12] presented a Retinex image enhancement algorithm based on the multigrid method, which first formulates the Retinex enhancement problem as a Poisson Equation and then uses the multigrid method to solve the equation efficiently. Zhang et al. [13] proposed a variational Retinex model based on global sparse gradient (GSG) guidance, which incorporates GSG Regularization to preserve the structural information and edge details of images, thereby improving the enhancement effect based on the Retinex principle. Song et al. [14] introduced an image enhancement method based on L0 regularization and re-weighted group sparsity (RGS), where L0 Regularization can better preserve image details, while RGS can better retain image colors.

In addition to image enhancement methods based on HE and Retinex theory, there are also other types of enhancement methods, such as those based on gradient field transformation [15–17]. The advantage of the gradient-based methods is that it can perform denoising while enhancing the image. However, the disadvantage of such methods is also evident, as the gradient domain method cannot achieve real-time image processing for large-sized images. This is because after enhancing in the gradient domain, it is necessary to convert back to the spatial domain, and currently there are two methods to achieve this process: matrix solving and gradient descent flow (GDF) methods [18]. However, both methods have relatively high computational resource requirements and are therefore time-consuming.

### 2.2. Deep Learning-Based Image Enhancement Methods

After neural networks were widely adopted in the field of Computer Vision, Lore et al. [19] first proposed a seven-layer sparse stacked autoencoder (SSAE) called LL-Net for improving the brightness of low-light image patches. However, the generalization

ability of this network was weak due to the use of artificially synthesized data as the training dataset. Subsequently, Wang et al. [20] proposed a low-light image enhancement network called GLADNet. This method firstly computes the global illumination estimation of the low-light image and then employs GLADNet to transform the low-light image into a high-illuminated image, guided by the global illumination estimation. Finally, the detail reconstruction of the enhanced image is achieved by fusing it with the original image. Lv et al. proposed [21] a multi-branch structure luminance enhancement network named MBLLEN. Cai et al. [22] proposed a single-frame image enhancement method, which includes two subnetworks for improving the brightness and details. Zhang et al. [23] presented a conditional re-enhancement network (CRENET), which can be combined with any type of image enhancement method to further enhance the contrast and brightness of the image based on the existing enhancement effect. All of the aforementioned methods are end-to-end supervised learning approaches. In addition, there exists another technical approach in the field of deep learning-based image enhancement, which combines the Retinex theory with neural networks. Wei et al. [24] first proposed to use the neural network called RetinexNet to decompose an image into an illumination map and a reflectance map. The method adjusts the brightness of the illumination map from a multi-scale perspective and removes the image noise from the reflectance map. Finally, the illumination and reflectance maps are pixel-wise multiplied to obtain the ultimate enhanced image. Following RetinexNet, Shen et al. proposed MSR-Net, which introduces the improved Retinex algorithm MSR as prior knowledge for low-light image enhancement [25]. Zhang et al. [26] introduced KinD, which builds upon RetinexNet by incorporating Adjustment-Net and Restoration-Net. These networks effectively correct illumination and reflectance images, leading to more accurate and plausible decomposition results.

For supervised learning methods, the cost of collecting paired datasets from the same scene is expensive. Additionally, artificially synthesized low-light images differ greatly from real images, and networks trained using such data often perform poorly on real data. As a result, some researchers have begun to turn their attention to unsupervised image enhancement methods. Unsupervised enhancement methods greatly reduce the algorithm's data requirements as they do not require paired images from the same scene during training. Jiang et al. [27] believe that the image enhancement problem is essentially an image style transfer problem and first proposed EnlightenGAN to achieve unsupervised image enhancement. Although EnlightenGAN reduces the network's demand for data, its model's enhancement performance is limited across different datasets. Yang et al. [28] contend that the primary limitation of EnlightenGAN is its relatively weak enhancement effect, leading them to propose a perceptual loss function that integrates Gamma correction. However, this modification failed to address the persisting issue of poor generalization across different datasets.In addition to the aforementioned methods, CycleGAN [29] and MUNIT [30] can also be used for unsupervised image enhancement tasks.

Compared to unsupervised learning, self-supervised image enhancement only requires low-light images as input without any reference images, further reducing the algorithm's data requirements. Zero-DCE proposed by Guo et al. [31] is a truly self-supervised image enhancement method. The essence of Zero-DCE is to utilize neural networks to fit the brightness enhancement curves, and then generate the brightness-enhanced image based on the curve and the original image. Zhang et al. [32] proposed maximizing the entropy of the channel with the maximum reflection component as a self-supervised enhancement constraint, and introduced the Contrast Enhancement Network (ICE-Net) and the Enhancement Denoising Network (RED-Net) for contrast enhancement and denoising, respectively. Ma et al. [33] proposed a self-supervised enhancement framework based on the Retinex illumination estimation principle. The method utilizes multi-module learning with shared weights and achieves favorable results on the basis of a lightweight network.

*2.3. Contribution of This Paper*

The following three ideas constitute our primary contributions to this field:

- We propose a novel attention network, MBCMHSA-Net, which combines multi-branch convolutional layers and a multi-head self-attention mechanism. This network use multiple branches of convolutional kernels to extract image features at different scales, and integrates contextual information using a multi-head self-attention mechanism. The MBCMHAS-Net has a simple structure that can be flexibly embedded into various branches of the backbone network. Our ablation and comparative experiments demonstrate that this module effectively enhances the detail features of images.
- We propose an attention map network that can dynamically generate attention maps, which overcomes the drawbacks of using fixed-pattern generated feature maps and improves the network's generalization ability. During the training process, the backbone network can pay more attention to important information in the image through the attention map, thereby enhancing the overall performance.
- We propose a novel discriminator network architecture that combines convolutional and Transformer encoders, and incorporates the MBCMHAS-Net proposed in this paper between convolutional layers. Compared with the convolutional neural network-based discriminator architecture, the proposed discriminator in this paper has better learning ability and can ensure a more stable training process.

## 3. Method

*3.1. Generator*

3.1.1. Generator Backbone Network

In the field of deep learning, the backbone network serves as the core component of deep learning models and is crucial for achieving machine vision tasks. Typically, a mature and stable network architecture or a pretrained model is used for the backbone network. Currently, commonly used backbone networks include VGG, ResNet, and others. For different vision tasks, it is necessary to select different backbone network structures. For example, in object detection tasks, some detection algorithms use ResNet-50 and Darknet-53 as backbone networks to extract image features, while in image generation tasks, convolutional autoencoders [19] and fully convolutional neural networks [34] are more commonly used. In this study, we adopt U-Net as the backbone network due to our experimental findings that, compared to convolutional autoencoders, using U-Net as the backbone network can allow our method to converge more easily. The underlying reason for this phenomenon could be attributed to the cross-layer skip connections implemented in the U-Net architecture, which enable feature fusion across different depths and avoid information loss.

The structure of the backbone network can be observed from Figure 2. After feeding the low-light image into the generator, the encoder of the backbone network extracts image features layer by layer, while the decoder restores the spatial resolution of the feature maps using upsampling layers. Both the encoder and decoder of the generator have three sublayers, each of which contains batch normalization layers and uses the ReLU function as the activation function. The encoder layer consists of two convolutional layers and one max pooling layer. In the corresponding decoder layer, bilinear interpolation-based upsampling layers are used instead of transposed convolutional layers, which can cause "blocky" granularity and hinder detail generation. After upsampling, convolutional layers are employed to combine spatial and channel information of the image to generate the final enhanced image. Since the pixel value range of the input image to the generator is [0,1] after preprocessing, the Tanh activation function is applied to the last layer of the backbone network to transform the pixel values of the enhanced image to the same range, which can accelerate network convergence.

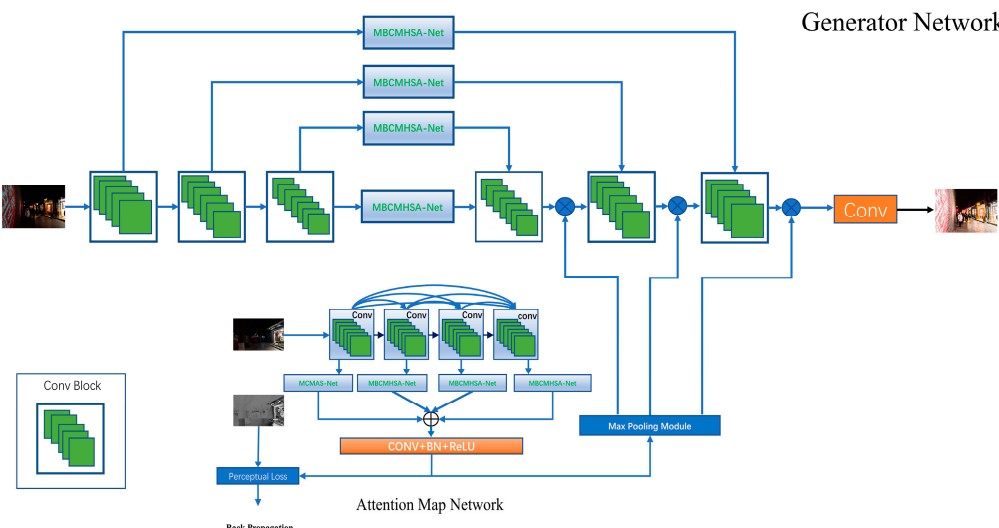

**Figure 2.** Network Structure of the Generator.

### 3.1.2. Attention Map Network

To avoid excessive brightness enhancement and improve the enhancement effect of image detail features, EnlightenGAN multiplies the attention map with the output of the U-Net decoder layer. The attention map used in EnlightenGAN is generated using a fixed pattern. The experimental results demonstrate that using the attention map can indeed accelerate the convergence speed of unsupervised networks, while removing it increases training time and results in inferior image enhancement. Based on this analysis, we conclude that the attention map plays an important role in improving image quality and network training. Compared with the EnlightenGAN, the attention map in our work is learned autonomously under the constraint of a single-channel attention map, and is multi-channel. Inspired by the channel attention mechanism and spatial attention mechanism of CBAM [35], we believe that a single-channel attention map only functions similarly to a spatial attention mechanism, while a multi-channel attention map can simultaneously consider both channel attention and spatial attention. Figure 3 depicts the network architecture of the attention map network.

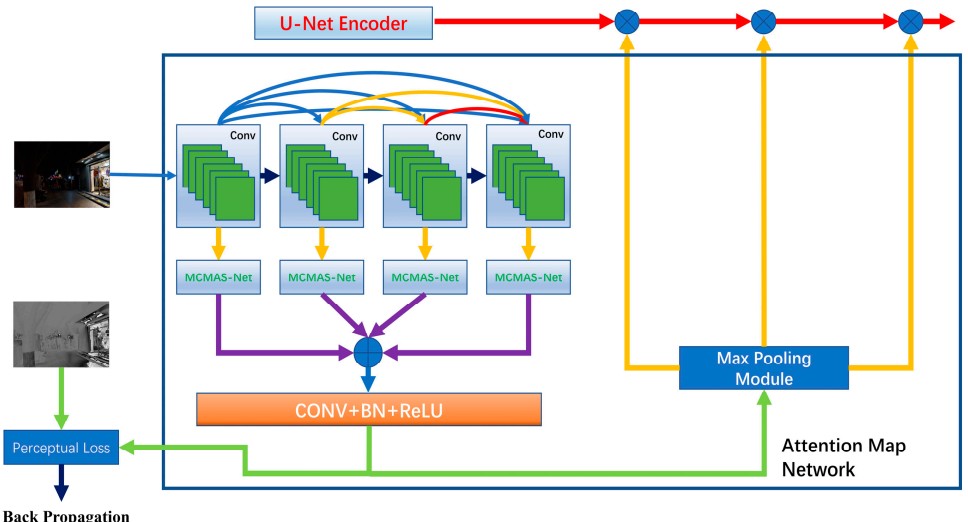

**Figure 3.** Network Structure of the Attention Map Network.

The proposed attention map network in this study consists of four densely connected convolutional modules. Following each convolutional module, a corresponding MBCMHSA-Net is attached to focus on features. All feature maps processed by the

MBCMHSA-Net are then channel-stacked and merged with feature and channel information via a combined convolutional module, which consists of a convolutional layer, a batch normalization layer, and a ReLU activation layer. Lastly, the attention maps are adjusted for channel and size by a max-pooling module, which consists of three parallel sub-layers, each comprising a convolutional layer, ReLU activation layer, and max-pooling layer. It can be observed from Figure 4 that the structure of this module is relatively straightforward.

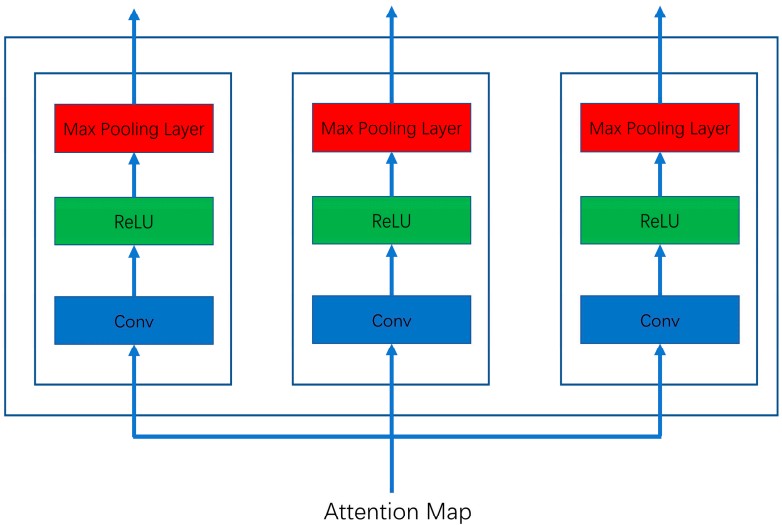

**Figure 4.** Network Structure of the Max Pooling Module.

### 3.1.3. Multi-Branch Convolutional Multi-Head Self-Attention Network

In deep learning networks, attention-based network or modules can improve image quality by assigning higher weights to important regions of the image. Additionally, in convolutional neural networks, the size of the convolutional kernel can affect the scale of the extracted features. Smaller kernels can capture more detailed features, while larger kernels can capture coarser features. For example, $1 \times 1$ kernels can be used for interchannel information interaction, $3 \times 3$ kernels can capture local features, and $5 \times 5$ or larger kernels can capture larger or global features. Based on these ideas, we propose a network that integrates multi-branch convolution and multi-head self-attention mechanisms, which we refer to as the Multi-Branch Convolution Multi-Head Self-Attention (MHSA) [36,37] Network (MBCMHSA-Net). This network is highly flexible and can be embedded into various branches of the main network. The structure of MBCMHSA-Net is shown in Figure 5.

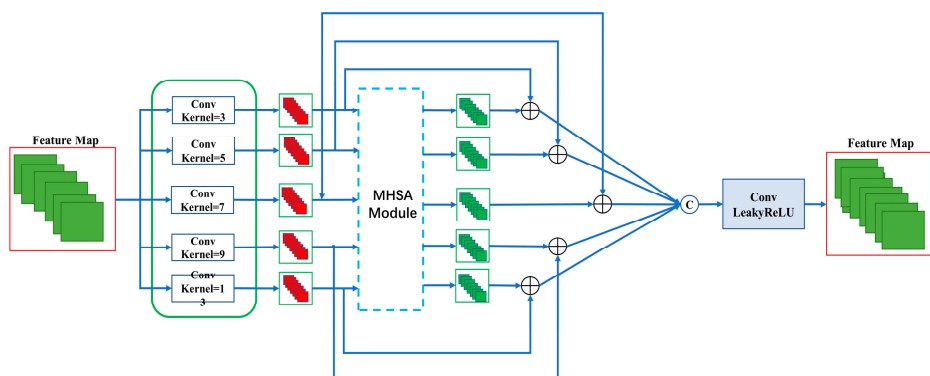

**Figure 5.** Network Structure of the MBCMHSA-Net.

In MBCMHAS-Net, we first set up five convolutional branches with filter sizes of $3 \times 3$, $5 \times 5$, $7 \times 7$, $11 \times 11$, and $13 \times 13$, to extract feature information at different scales. After feature extraction, the feature maps are fed into the MHSA module, and the weight maps generated by the MHSA module need to be pixel-wise added to the input feature

maps. Finally, all the feature maps processed by the weight maps are pixel-wise added and fed into a network layer for feature fusion, which consists of two convolutional layers and a LeakyReLU activation function. It should be noted that due to the excessive number of network parameters caused by large convolutional kernels, different channel numbers are set for different convolutional layers in this study, and the larger the kernel size, the fewer the number of feature map channels for that convolutional layer. In addition, it is not feasible to directly input multi-channel feature maps into MHSA, as this would also result in an excessively large dimension of the fully connected layer in the MHSA. Therefore, before processing the multi-branch feature maps in MHSA, a $1 \times 1$ convolutional layer is used to reduce the dimensions of the feature maps. Equations (1) and (2) describe the MHSA procedure:

$$Q, K, V = \text{Chun}k[C_{1 \times 1}(C_{3 \times 3 \sim 13 \times 13}(x))] \tag{1}$$

$$Attention(Q, K, V) = softmax(\frac{QK^{T}}{\sqrt{d_k}})V \tag{2}$$

Here, $1 \times 1$, $3 \times 3$, and so on represent the sizes of the convolution kernel; $C$ represents the convolution operation; and $Q$, $K$, and $V$ represent the word embedding vector matrix formed by the Query, Key, and Value in the self-attention mechanism, respectively. Finally, the feature map processed by MHSA should be processed by a network sublayer that contains two convolutional layers and a LeakyReLU activation layer.

### 3.2. Discriminator

The role of the discriminator in generative adversarial networks is to provide gradient descent direction for the training of the generator. In our experiments, we found that the CNN-based discriminator was inadequate in guiding the training of the generator, leading to unstable training and low-quality image generation. To address this issue, we propose an improved discriminator network, which consists of a convolutional encoder, a Transformer encoder, and a fully connected neural network. Specifically, the convolutional encoder is employed to extract local features of the images, the Transformer encoder further processes the features extracted by the CNN, thereby providing superior contextual information to the entire discriminator network, and the fully connected neural network is utilized to transform the output vector dimension of the Transformer.

Figure 6 delineates the specific architecture of the discriminator network, from which it can be discerned that image features are initially extracted via a convolutional encoder. Each convolutional module of the encoder consists of a convolutional layer, a batch normalization layer, a max pooling layer, and a ReLU activation layer. In addition, the MBCMHSA-Net, proposed in this study, is embedded between the blocks of the convolutional encoder to focus on features. As the image passes through the convolutional encoder, the size of its feature maps gradually reduces, and after the last convolutional module, we obtain the final feature map from the feature extraction stage.

The feature maps obtained from the convolutional encoder cannot be directly fed into the Transformer encoder and require preprocessing. This preprocessing involves patch embedding and position embedding. Prior to embedding, the feature maps are first partitioned into blocks, then each block image is stretched into a one-dimensional vector and subjected to linear transformation to reduce the dimension of the vector. Finally, position encoding is applied to the vectors to obtain word embedding vectors. After obtaining the embedding vectors, they are fed into the Transformer encoder, which consists of three Transformer encoder layers. To improve network performance, residual connections are established between the input and output of the encoder in this study.

Figure 7 illustrates the loss trend of the generator and discriminator during the training process in this study. It can be observed from the figure that the loss value of the generator shows a fluctuating trend of initial increase followed by a decrease, while that of the discriminator steadily decreases and then fluctuates within a small range. This indicates the adversarial relationship between the generator and the discriminator. Additionally, the

figure shows that the discriminator has a relatively strong learning ability, which is consistent with the current training strategy for generative adversarial networks. Specifically, a robust discriminator is required to provide effective guidance for the generator during the training process of the generative adversarial network.

Discriminator Network

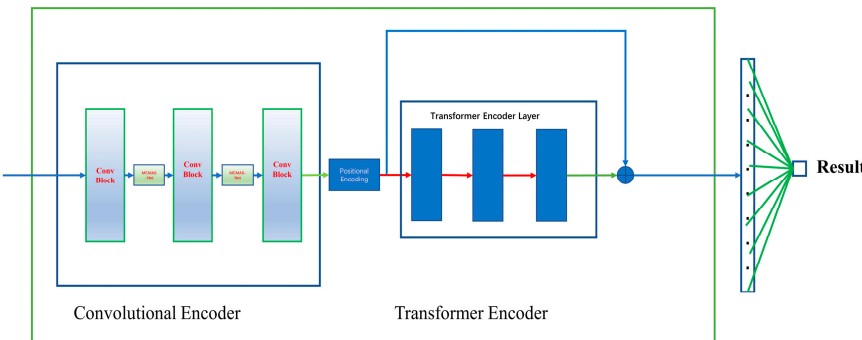

**Figure 6.** The Network Structure of the Discriminator.

The feature maps obtained from the convolutional encoder cannot be directly fed into the Transformer encoder and require preprocessing. This preprocessing involves patch embedding and position embedding. Prior to embedding, the feature maps are first partitioned into blocks, then each block image is stretched into a one-dimensional vector and subjected to linear transformation to reduce the dimension of the vector. Finally, position encoding is applied to the vectors to obtain word embedding vectors. After obtaining the embedding vectors, they are fed into the Transformer encoder, which consists of three Transformer encoder layers. To improve network performance, residual connections are established between the input and output of the encoder in this study.

Figure 7 illustrates the loss trend of the generator and discriminator during the training process in this study. It can be observed from the figure that the loss value of the generator shows a fluctuating trend of initial increase followed by a decrease, while that of the discriminator steadily decreases and then fluctuates within a small range. This indicates the adversarial relationship between the generator and the discriminator. Additionally, the figure shows that the discriminator has a relatively strong learning ability, which is consistent with the current training strategy for generative adversarial networks. Specifically, a robust discriminator is required to provide effective guidance for the generator during the training process of the generative adversarial network.

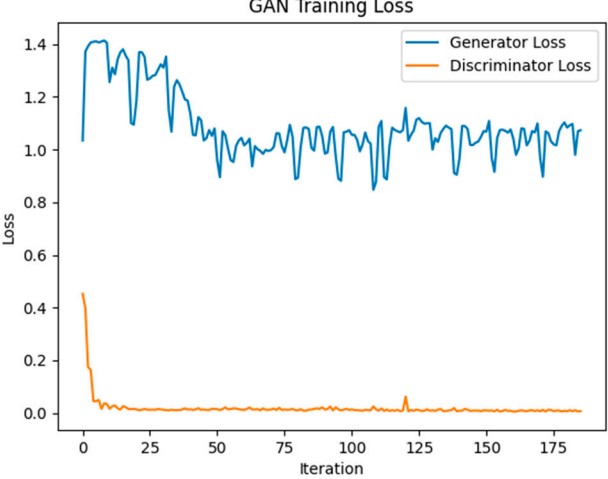

**Figure 7.** The Loss Trend of improved Discriminator Networks and Generator.

*3.3. Loss Function*

The loss function mainly constitutes three terms:

- **Perceptual Loss:** This function is utilized to evaluate the content loss between the generated image and the input image, as well as the generated attention map and the original attention map. In the experiment, we first obtain the convolutional feature maps named "Conv1-1", "Conv2-1", "Conv3-1", and "Conv4-1" for both the low-light image and the enhanced image from the pre-trained VGG19 network. Then, we utilize the L2 loss function to measure the difference between these two feature maps. When evaluating the attention map, we convert the single-channel attention map into a three-channel image and then compute the perceptual loss between it and the generated attention map. The specific calculation process of this loss is as follows:

$$L_{perceptual} = \frac{1}{C_j H_j W_J} \|\phi_j(y) - \phi_j(\dot{y})\|^2 \tag{3}$$

where $C_j \times H_j \times W_j$ represents the size of the feature map, $\phi_j(y)$ represents the low-light image, and $\phi_j(\dot{y})$ represents the enhanced image.

- **Adversarial Loss**: The logarithmic form of adversarial loss is used by the traditional GAN. However, this adversarial loss has the issue of unstable training, particularly when using the sigmoid activation function in the discriminator's final layer, which may result in the issue of gradient disappearance. Therefore, we use the least squares adversarial loss (LSGAN) [30] function to train our model. The form of LSGAN loss is expressed in Equations (4) and (5):

$$L_D = E_{X_r \sim P_{real}}[(D_{R_a}(X_r, X_f) - 1)^2] + E_{X_r \sim P_{fake}}[(D_{R_a}(X_r, X_f))^2] \tag{4}$$

$$L_G = E_{X_r \sim P_{real}}[(D_{R_a}(X_r, X_f) - 1)^2] \tag{5}$$

- **TV-Loss**: We use the total variation loss function to limit the image noise. The total variation loss function is widely used for removing image noise while maintaining the image edge information.

$$L_{TV} = \|\nabla Y\|_1 \tag{6}$$

where $\nabla Y$ represents the gradient of the image.

The overall loss function of the network is:

$$\text{Loss} = L_{perception} + L_{adversial} + L_{TV} \tag{7}$$

## 4. Experiment

The experiment in this study involves four parts: experiments on image enhancement under low-light and extremely low-light conditions, image application experiments, and network ablation experiments. The experiment on low-light image enhancement is a relatively conventional one, which mainly evaluates the image quality of different algorithms for enhancing dim images, as most enhancement algorithms may be effective in this environment. The experiment on extremely low-light image enhancement mainly examines the algorithmic robustness under such conditions, since images captured in extremely low-light conditions contain a large amount of image noise, and some methods may not work properly, such as the BPDHE algorithm. The application experiment mainly evaluates the effect of the proposed method on low-light image matching technology, which can indicate whether the method can provide assistance for other visual tasks in addition to improving visual effects. The network ablation experiment primarily assesses the effectiveness of the proposed network.

*4.1. Experiment Details*

4.1.1. Dataset

The datasets DarkFace [38] and LSRW [39] are used for training of Generative Adversial Networks. Both the DarkFace and LSRW datasets are nighttime image datasets collected in the natural environment, where the DarkFace training set contains 6000 images with a 1080 × 720 resolution, and these data have no corresponding normal light label image, while the LSRW (Nikon) contains 3150 pairs of "low-normal" paired images with a 960 × 640 resolution. The Dark Face dataset was captured by researchers using Sony cameras. The dataset primarily consists of dark streets captured during the shooting process, with the camera's exposure time constantly changing. The longest exposure time is 1 s, making the images in this dataset fall within the category of low-light images. The LSRW dataset consists entirely of images captured by Huawei smartphones and Nikon cameras. The Nikon cameras were set to exposure times of 0.00 s and 0.5 s, with ISO values of 50 and 100, and a total of 3170 pairs of images were captured. The Huawei smartphones were set to an exposure time of 0.0025 s and an ISO value of 50, and a total of 2480 pairs of images were captured. Tripods were used to fix the cameras throughout the entire shooting process, and all images containing moving objects were removed during data processing.

We selected 600 low-light images and 850 images from the Dark Face dataset and LSRW dataset as the unsupervised dataset. These images do not have a corresponding relationship, so we can realize unsupervised training. Additionally, we also selected 700 pairs of paired images from the LSRW dataset as the supervised learning dataset. In order to test the performance of the network in extremely low-light environments, we selected the SID [40] dataset as the validation dataset. The SID dataset contains long-exposure and multi-frame short-exposure images of the same scene. The exposure time during the collection of this dataset was 0.04 s and 0.1 s. In the application experiment, we use the synthetic image dataset of LoL [24]. We resized all images to 540 × 360 for training.

It should be noted that the method proposed in this study employs two modes of training throughout the entire testing process. One mode is supervised learning training based on the LSRW dataset, while the other is unsupervised learning using both the Dark Face and LSRW datasets. Specifically, the supervised learning image enhancement methods involved in the experiments are trained using the same LSRW dataset as ours, while the unsupervised and self-supervised methods are trained using the combined unsupervised dataset and the Dark Face dataset, respectively. The neural networks of all the deep learning enhancement methods are trained for 50 epochs for comparison.

4.1.2. Training Details

The experimental code in this study was implemented using Pytroch 1.8.0. The entire algorithm can be easily duplicated because the initialization weights and biases of each network layer are completed by default in accordance with the Pytorch framework, and the random seed value is not set. During the training phase, we used the Adam optimizer, set the batch size to 8, and set the learning rates to 0.0001 for both the generator and discriminator. The values for Adam are 0.9 and 0.999, respectively. The overall network training process was fairly stable owing to the adversarial loss function of the LSGAN. Additionally, we used an NVIDIA Tesla m40 GPU with 24 GB of memory.

*4.2. Objective Assessment*

4.2.1. Low-Light Image Enhancement Experiment

In comparison experiments, this study introduced six enhancement methods based on the learning process (GLADNet, KinD, MBLLEN, RetinexNet, SCI, and Zero-DCE) and three traditional image enhancement methods (LIME [41], MF [42], and SIRE [43]).

Subjective Evaluation of Image Quality

Figure 8 demonstrates the enhancement effects of the generator. It can be observed from Test Image 1 that the MBLLEN and SIRE methods exhibited the issue of over-

enhancement in certain image regions. The enhancement in these areas was too strong, which resulted in the obscuring of original contours and details. The image quality of the RetinexNet images showed an unnatural color tone, which significantly differed from that of natural images. This difference in color tone can be easily perceived by human eyes. Among the tested methods, LIME, MF, and most deep learning-based methods exhibited good performance. In Test Image 2, SIRE significantly increased the brightness of the image. GLADNet, KinD, and EnlightenGAN produced images with high contrast, while the proposed method appeared to have slightly lower contrast and higher brightness. In comparison, the proposed method significantly enhanced image brightness without producing adverse results such as over-enhancement, shadows, or color distortion.

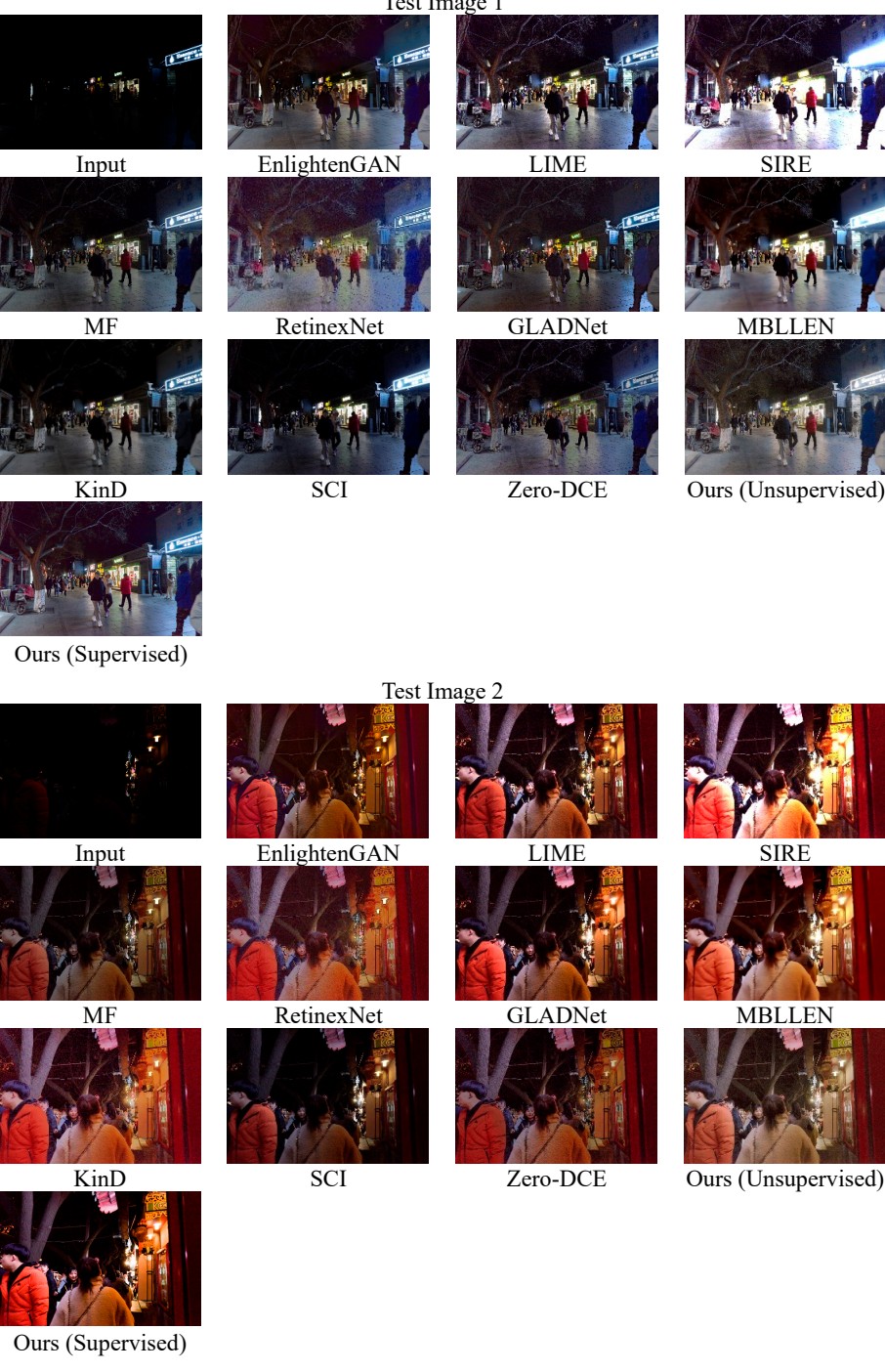

**Figure 8.** Comparison of the Effects in Low-Light Conditions.

Objective Evaluation of Image Quality

Subjectively evaluating image quality based solely on human visual perception is not rigorous enough. Therefore, we adopted currently prevalent image quality metrics for objective evaluation. These metrics include NIQE, CEIQ, LOE, entropy, and standard deviation (SD). The following provides a brief explanation of each metric.

NIQE: This is a commonly used unsupervised image quality evaluation metric. It can assess the quality of digital images with statistical models. A lower NIQE score indicates better image quality.

CEIQ: This metric evaluates the enhancement effect by calculating the differences in factors such as structural similarity, color fidelity, and contrast between the enhanced image and the original image. A higher CEIQ value indicates better image quality.

LOE: This metric is specifically designed for the field of low-light image enhancement. It is mainly used to measure the algorithm's ability to maintain brightness order, which is essentially evaluating the difference in brightness distribution before and after image enhancement.

Entropy: This metric is mainly used to measure the diversity of pixel value distribution in an image. The magnitude of the entropy metric reflects the richness of the image information. A larger entropy value indicates greater information content in the image, while a smaller entropy value indicates less information. In an 8-bit unsigned image, the maximum value of entropy is 8.

SD: SD (standard deviation) is a statistics-based image quality evaluation metric. The larger the SD value, the clearer the image, while a smaller SD value indicates that the image details are more blurry.

After training the network for 30 epochs, we selected the first test image in Figure 8 to calculate the metric values and compared them in Table 1.

**Table 1.** Different Image Metrics in Low-Light Conditions.

| Method | NIQE | CEIQ | LOE | EN | SD |
|---|---|---|---|---|---|
| Input | 4.450 | 1.89 | — | 4.36 | 25.65 |
| EnlightenGAN | 2.613 | 3.14 | 527.54 | 7.18 | 40.34 |
| GLADNet | 2.364 | 5.38 | 188.31 | 7.35 | 45.15 |
| KinD | 2.549 | 5.23 | 256.05 | 7.28 | 43.49 |
| LIME | 2.774 | 6.54 | 339.88 | 7.69 | 59.79 |
| MBLLEN | 2.994 | 5.13 | 162.27 | 7.32 | 52.02 |
| SCI | 2.721 | 4.18 | 189.59 | 7.19 | 55.92 |
| MF | 2.578 | 4.49 | 436.01 | 6.97 | 36.93 |
| SIRE | 3.418 | 4.18 | 420.98 | 6.23 | 87.42 |
| RetinexNet | 4.036 | 5.38 | 490.51 | 7.21 | 41.60 |
| Zero-DCE | 2.732 | 4.84 | 215.18 | 7.14 | 41.48 |
| Ours (Supervised) | 2.513 | 5.26 | 204.76 | 7.76 | 76.30 |
| Ours (Unsupervised) | 2.494 | 5.21 | 126.34 | 7.34 | 51.06 |

From the data in the Table 1, it could be seen that the LIME method performed excellently in terms of contrast and color fidelity, while also exhibiting high values of standard deviation. This suggested that images enhanced by LIME had clearer details. The algorithm RetinexNet performed the worst in the NIQE metric. As shown in Figure 8, the images enhanced by RetinexNet suffered from color distortion and exhibited significant differences from real images in terms of image style. The best method for the NIQE metric was GLADNet. Although not as bright as the LIME and MF methods, GLADNet produced images with complete details and without over-enhancement. In this study, the images trained via an unsupervised method performed better in terms of LOE and NIQE metrics, indicating that the brightness distribution of the enhanced images was more similar to the original images. The images trained via a supervised method performed best in terms of the information entropy metric, and were second only to the LIME method in terms of the standard deviation metric, indicating that the image details were clearer than the

other methods. The worst methods for the LOE metric were RetinexNet, MF, and SIRE, indicating that the light distribution of the images produced by these three methods differed greatly from the original image. Among these, SIRE had the most severe over-enhancement problem, resulting in overexposed areas in the image. Overall, our proposed method exhibited outstanding performance in all metrics, while EnlightenGAN had a relatively average performance in terms of metrics.

### 4.2.2. Extreme Low-Light Image Enhancement Experiment

The test images selected in this section were sourced from the SID dataset, which contains images with illumination levels ranging from 0.2 lux to 5 lux, and hence satisfies the definition of extremely low-light images. In this study, an extreme low-light image was randomly selected from the SID dataset for testing. As the image data provided in the SID dataset are large-scale RAW format images, we converted the selected image using bilinear interpolation to a $540 \times 360$ JPG format image for testing purposes.

### Subjective Evaluation of Image Quality

Among the tested image enhancement methods, including Zero-DCE, KinD, EnlightenGAN, and SCI, the improvements in brightness and contrast were relatively limited. The RetinexNet method still produced significant changes to the style and tone of the image. Among these ten contrast methods, MBLLEN, LIME, MF, and our method showed more prominent visual effects. The MBLLEN and LIME methods significantly improved the brightness and contrast of the image, resulting in more vivid colors. However, MBLLEN-enhanced images did not contain noise or mosaic artifacts compared to LIME, indicating its superior denoising effect. The advantages of LIME and MF lie in generating more vibrant colors. When employing both unsupervised and supervised training, the method proposed in this study demonstrated a significant difference in enhancing extremely low-light images. This difference can be observed in Test Image 2. Specifically, if the proposed method is trained using supervised learning, it can significantly improve the brightness and contrast of the images. On the other hand, if the method is trained using unsupervised learning, the resulting images have lower brightness. However, it still has a significant improvement effect compared to the other methods. The method proposed in this study can improved the visual quality of images. However, compared to the MBLLEN method, our approach still has limitations in suppressing image noise.

### Subjective Evaluation of Image Quality

This section selects the second test image in Figure 9 for performance evaluation. The image quality evaluation metrics discussed in this section are the same as those in the previous section. Table 2 displays the metric values of the enhanced images obtained using the different methods.

As shown in Table 2, the MBLLEN performed better on metrics such as NIQE, while our method performed the best on CEIQ and LOE metrics. Among all metrics, RetinexNet performed the worst on NIQE and CEIQ, which mainly reflected the naturalness of the image. The results in Figure 9 confirmed this observation. Regarding the LOE metric, EnlightenGAN and SCI performed poorly, indicating their limited ability to enhance image brightness, and the significant difference between the brightness distribution of the enhanced images and the original images suggested a problem with the generalization ability of these two networks. Furthermore, this also illustrates the drawbacks of employing a pre-determined pattern to generate attention maps. Our method (unsupervised) and MBLLEN achieved the best performance on the entropy metric, while Zero-DCE and SCI performed poorly. Regarding the poor performance of Zero-DCE in extremely low-light testing, we believed this may have been due to the loss function of Zero-DCE, which imposed excessive constraints on the brightness distribution of the image during network training to avoid excessive brightness enhancement. However, the threshold used for brightness enhancement in Zero-DCE was limited in its effectiveness. Overall, based on the comprehensive

evaluation, we concluded that our proposed method and MBLLEN achieved the most significant improvement in image quality in extremely low-light situations.

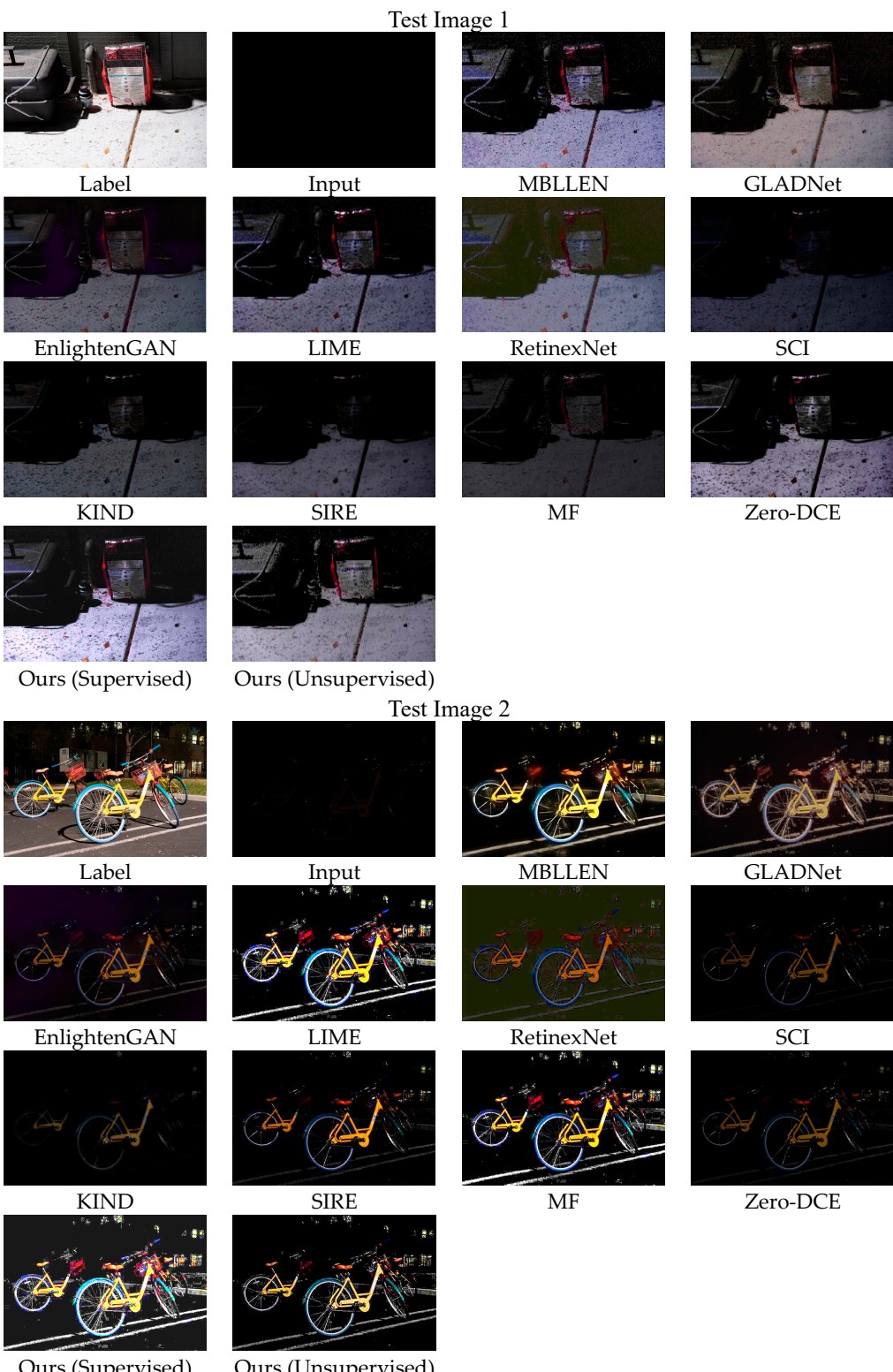

**Figure 9.** Comparison of the Effects of Different Algorithms in in Extremely Low-Light Conditions.

**Table 2.** Different Image Metrics in Extremely Low-Light Conditions.

| Method | NIQE | CEIQ | LOE | EN | SD |
|---|---|---|---|---|---|
| Input | 7.145 | 1.41 | — | 1.42 | 3.56 |
| EnlightenGAN | 6.203 | 1.86 | 1227.99 | 4.72 | 19.00 |
| GLADNet | 4.929 | 2.19 | 891.95 | 4.58 | 39.71 |
| KinD | 6.612 | 1.53 | 799.48 | 3.06 | 11.48 |
| LIME | 8.053 | 1.99 | 511.69 | 3.65 | 67.23 |
| MBLLEN | 4.296 | 2.24 | 876.95 | 5.63 | 58.74 |
| SCI | 8.431 | 1.85 | 1264.53 | 1.94 | 13.71 |
| MF | 7.653 | 1.96 | 522.89 | 3.61 | 61.11 |
| SIRE | 7.328 | 1.93 | 475.48 | 3.17 | 30.72 |
| RetinexNet | 10.951 | 1.35 | 528.25 | 4.16 | 17.14 |
| Zero-DCE | 6.842 | 1.85 | 368.61 | 2.72 | 16.08 |
| Ours (Supervised) | 5.803 | 1.88 | 587.64 | 4.29 | 65.19 |
| Ours (Unsupervised) | 4.579 | 2.03 | 353.69 | 5.41 | 52.85 |

### 4.3. Ablation Experiment

It can be observed from Figure 10 that using only the backbone network without incorporating any additional modules, the images generated by the generator exhibited preliminary enhancement effects, but with low contrast, which gives a pronounced sense of granularity and certain features appeared blurred. However, if the Attention Map Network is removed and only the backbone network and MBCMHSA-Net are retained, the image will exhibit a good enhancement effect, but there may be slight block of artifacts in some cases. According to our analysis, the preprocessing of the image is the cause of the block phenomenon. Every image must be evenly divided into multiple patches during the specific implementation process when using the multi-head self-attention mechanism. Patch boundaries may be generated because every patch must pass through the embedding layer and the linear layer. These boundaries gradually vanish as the training period is extended, even though this method creates relatively distinct boundaries between different portions of the image. According to our tests and experiments, the minimum number of training epochs required to eliminate the block boundary was 400. The result of removing the attention module is the effect shown in Column 4 of the ablation experiment. An overall improvement was only achieved using the attention map network in the early stages of training. The image gradually became clearer as the number of training epochs increases, although the details were initially hazy. The final effect of our algorithm is shown in the last column of Figure 10. We can see that the final result we obtained both avoids the blocky effect of the image and had a good enhancement effect because our generator network combines the advantages of the attention modules and attention map networks. The attention map network was responsible for improving the overall brightness of the image, and the attention module enhanced the detailed features of the image. The attention map network maintained the relationship between the features of each patch of the image during the learning process.

Our method can achieve a preliminary enhancement effect in the first epoch. Although the tone may vary, the details are already relatively complete. Note that each convolutional layer of the generator and discriminator in this study adopts a $5 \times 5$ convolution kernel, and the use of a $3 \times 3$ convolution kernel significantly reduces the speed of network convergence. However, this phenomenon is not unrestricted, as attempting larger convolution kernels did not lead to improved results in the network. Although the attention map network can speed up convergence and eliminate the image-blocking effect, it also leads to an improvement in the overall brightness of the image. This enhancement effect is sometimes unnatural. As shown in the first row of Figure 10, the brightness enhancement effect in the image surpassed the contrast enhancement effect. We suspect that this is because the enhancement effect of the attention map network was too strong, causing the attention module effect to be masked.

| Input | Backbone Only | Without Attention Map Network | Without Attention Module | Full Network |
|---|---|---|---|---|

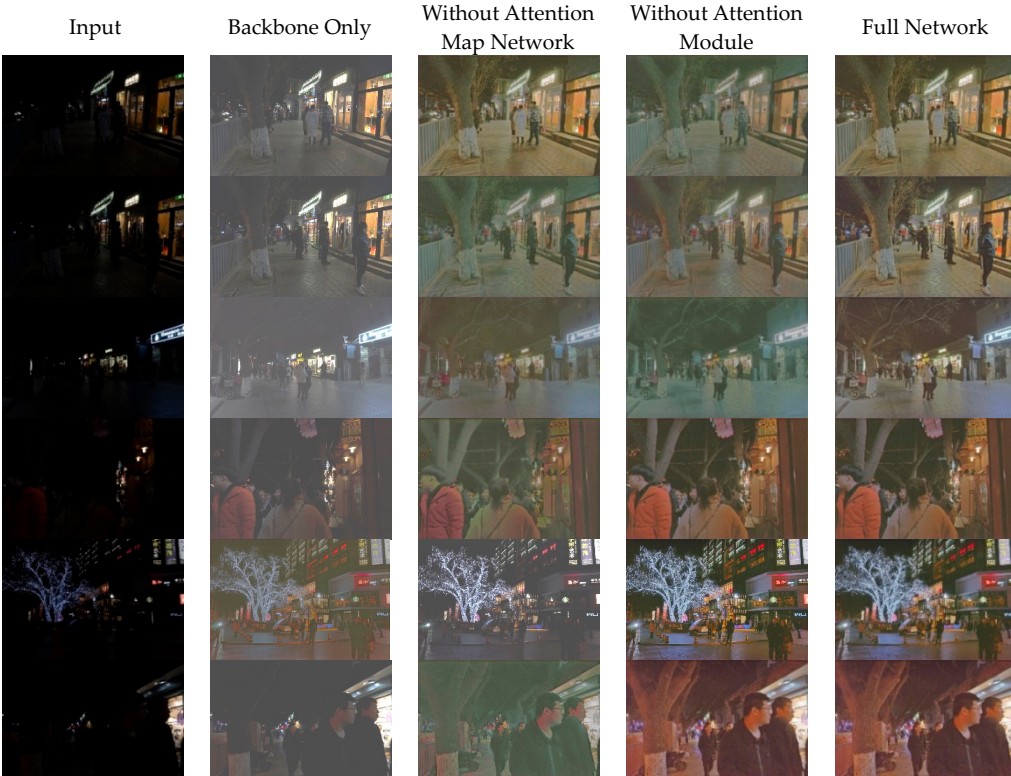

**Figure 10.** Ablation Experiment Images.

### 4.4. Application Experiment: Low-Light and Normal-Light Image Matching

This study introduced an application-oriented experiment aimed at comparing the application value of various methods. As there is widespread application of low-light image enhancement in various fields, this experiment utilized low-light image matching as the test task. Image matching is a common technology in industrial image processing. It plays an important role in target tracking, image target localization, and multiple image target detection. The corresponding image-matching algorithms are unable to achieve good matching results in low light, despite being widely employed in several machine vision fields. The most effective way to handle this issue is to enhance the low-light image before matching the image. We use the Sift operator based on feature point detection for specific image matching. Sift is an operator for the local description of image features based on a scale space that is invariant to scaling, rotation, and even affine transformation of the image. The specific steps of the algorithm are to extract key points from the paired images, then attach detailed information (local features) to the key points, and finally find multiple matching feature points by comparing the two feature points.

The application experiment consisted of three parts: low-light image matching, extremely low-light image matching, and medium-light image matching. The medium-light image refers to images with non-uniform lighting, where the illuminance in some areas of the image are limited, while others are overexposed. The test images for the low-light, extremely low-light, and medium-light testing come from the LoL dataset, the SID dataset, and the SCIE dataset, respectively. The SCIE dataset contains 4413 pairs of multi-exposure images. The author collected image sequences with over 10,000 different exposure times using seven types of cameras. Additionally, the reference images in this dataset were generated using the MEF HDR algorithm. The procedure of each experiment was as follows:

(1) Randomly select a pair of 'low-normal' images captured in the same scene from the LoL dataset.
(2) Computed the matching results between the low-light image and the normal-light image, then recorded the results.
(3) Enhanced the low-light images using the different image enhancement methods.

(4)   Computed the matching results between the enhanced images and the normal-light image.

(5)   Compared the matching results and determined which method yielded the greatest improvement in low-light image matching.

4.4.1. Image Matching Experiment in Low-Light Conditions

In image matching algorithms, both the matching rate and the number of detected points are crucial factors. The matching rate refers to the ratio of the number of matched feature points to the total number of feature points in two images. A higher matching rate indicates greater accuracy in matching, but if the number of feature points in the two images differs significantly, the matching rate may lose its meaning.

Figure 11 shows visualization of image matching results enhanced by different algorithms. Table 3 shows the matching results of images enhanced by various algorithms. In this table, different algorithms correspond to different numbers of feature points and matching points, and it can be observed that there was a significant difference in the number of these points. The first row of Table 3 presents the results of matching the original image without any enhancement processing and a reference image with high brightness. The fact that there were 0 feature points indicates that the SIFT algorithm cannot perform image matching in low-light conditions. The second row in Table 3 presents the results of matching two identical high-brightness reference images, and it can be observed that the quality of these feature points was high, and the matching rate reached 100%. The results in the first and second rows represent the worst and best cases of image matching in low-light conditions, respectively. We used the second result as a standard for algorithm comparison.

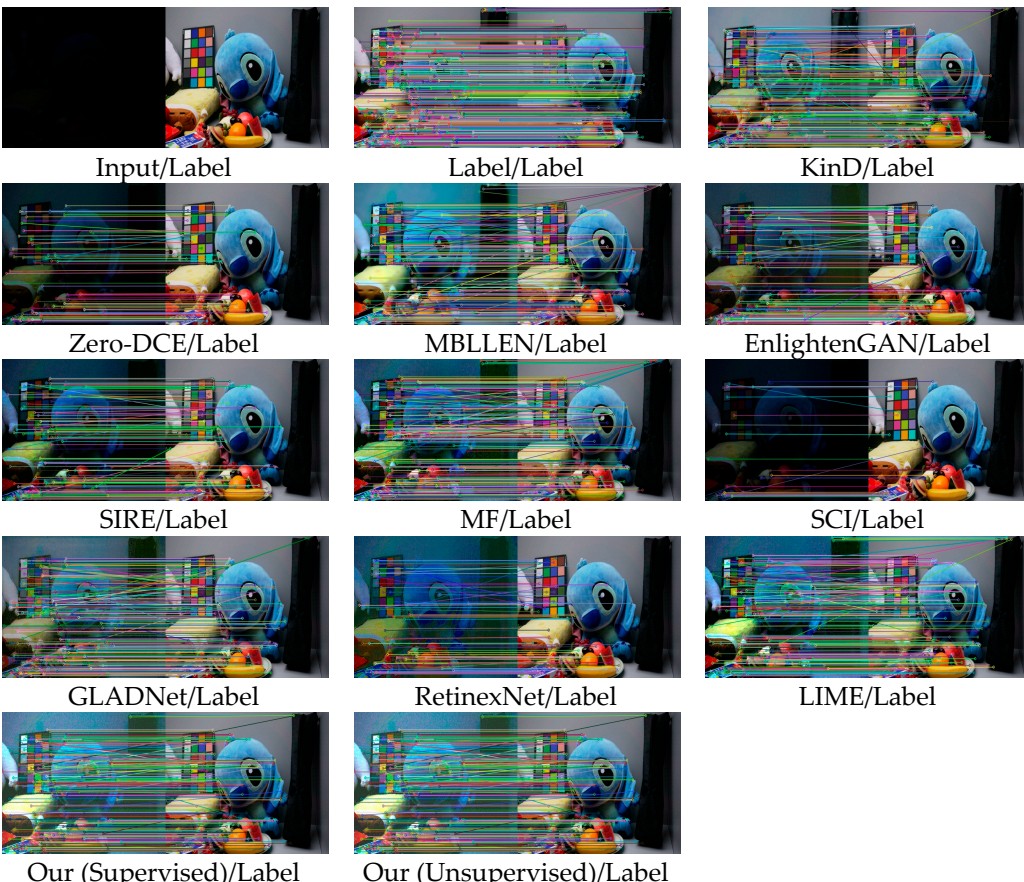

**Figure 11.** Visualization of image matching results enhanced by different algorithms.

**Table 3.** Comparison of image matching results enhanced by different algorithms.

| Match Mode | Feature Points | Match Points | Match Rate |
|---|---|---|---|
| Input/Label | 0 | 0 | 0 |
| Label/Label | 739 | 739 | 100.0% |
| KinD/Label | 769 | 278 | 36.2% |
| Zero-DCE/Label | 181 | 88 | 48.6% |
| MBLLEN/Label | 548 | 269 | 49.1% |
| EnlightenGAN/Label | 364 | 143 | 39.3% |
| SIRE/Label | 440 | 157 | 35.6% |
| MF/Label | 789 | 247 | 31.3% |
| SCI/Label | 80 | 29 | 36.3% |
| GLADNet/Label | 852 | 258 | 30.3% |
| RetinexNet/Label | 506 | 91 | 17.9% |
| LIME/Label | 859 | 258 | 30.0% |
| Our (Supervised)/Label | 549 | 283 | 51.5% |
| Our (Unsupervised)/Label | 515 | 257 | 49.9% |

Among all the algorithms, the LIME algorithm corresponded to the largest number of feature points, but the number of matching points was relatively small. This suggests that the quality of the feature points extracted using the SIFT algorithm was poor, leading to a lower matching rate. Except for the LIME algorithm, the MF, LIME, and GLADNet also suffered from this issue. After analysis, we believe that this may be due to image blurring or excessive noise enhancement caused by image enhancement algorithms. To verify this hypothesis, we estimated the overall noise level of the enhanced image using the local variance noise estimation method. The results in Table 4 indicated that the noise levels of the LIME and MF methods were significantly higher than those of the other algorithms, which confirmed our hypothesis. In addition, EnlightenGAN, SCI, and SIRE had a relatively average effect on improving image matching. Overall, our method (supervised) had the best effect on low-light image matching, followed by our proposed unsupervised learning-based method and the MBLLEN method.

**Table 4.** Comparison of Image Noise Level of Different Algorithms.

| Image | Local Variance Noise Estimation |
|---|---|
| Input | 0.0000097 |
| Label | 0.00248 |
| KinD | 0.00191 |
| Zero-DCE | 0.00103 |
| MBLLEN | 0.00061 |
| EnlightenGAN | 0.00055 |
| SIRE | 0.00103 |
| MF | 0.00904 |
| SCI | 0.00021 |
| GLADNet | 0.00669 |
| RetinexNet | 0.00715 |
| LIME | 0.01432 |
| Our (Supervised) | 0.00189 |
| Our (Unsupervised) | 0.00204 |

4.4.2. Image Matching Experiment in Extremely Low-Light Conditions

From Figure 12 and Table 5, it can be observed that under extremely low-light conditions, the performance gain of various enhancement algorithms on image matching results had decreased. Specifically, LIME, MF, and GLADNet had significantly increased the number of feature points, but the matching rate dropped significantly, indicating that the images enhanced by these three algorithms had excessively high levels of noise, which can be seen from Table 6. Although other methods such as EnlightenGAN, Zero-DCE, and

KinD had only average enhancement effects on the images, their advantage is that the noise level of the images was not significantly increased, which may explain why these methods with only average enhancement effects had relatively higher matching rates. To compare the image matching results and noise levels under low-light and extremely low-light conditions, we considered noise as only one of the factors that affects the matching accuracy. When the noise level is too high, it may mislead image matching algorithms such as SIFT and produce a large number of feature points, but whether these feature points meet the conditions of image matching still depends on the quality of the image features themselves. Under extremely low-light conditions, the information in the image is limited, and a large amount of noise is generated in the image. Therefore, we believe that removing noise and effectively enhancing features such as edges and contours in the image are the key to improving the image matching results. The proposed method in this study had higher matching rates and more matched feature points compared to the other algorithms under both supervised and unsupervised learning, indicating that it performs better in improving image matching rates under extremely low-light conditions compared to SIRE and the other methods.

### 4.4.3. Image Matching Experiment in Medium-Light Condition

In the third section of the application experiment in this study, image matching is mainly performed on images with uneven illumination. These types of images are characterized by an overall medium level of brightness, with some areas having high brightness and other areas possibly being underexposed. Conventional image enhancement methods, such as histogram equalization, can easily cause the originally bright areas of the image to become even brighter. This is the image over-enhancement problem that our method focused on solving. Over-enhancement of image brightness, similar to over-enhancement of noise, can have a negative impact on image matching and result in overexposure.

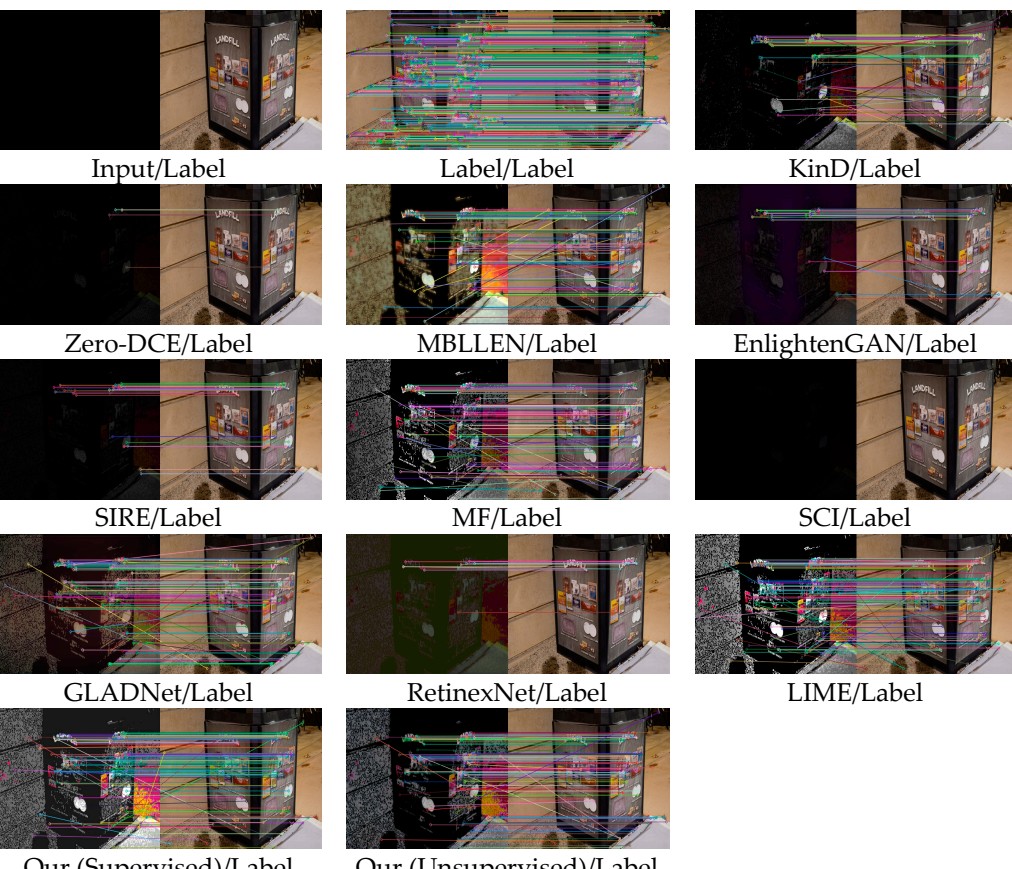

**Figure 12.** Visualization of image matching results enhanced by different algorithms.

**Table 5.** Comparison of image matching results enhanced by different algorithms.

| Match Mode | Feature Points | Match Points | Match Rate |
|---|---|---|---|
| Input/Label | 0 | 0 | 0 |
| Label/Label | 870 | 870 | 100% |
| KinD/Label | 661 | 73 | 11.0% |
| Zero-DCE/Label | 11 | 6 | 54.5% |
| MBLLEN/Label | 1050 | 87 | 8.3% |
| EnlightenGAN/Label | 97 | 36 | 37.1% |
| SIRE/Label | 65 | 31 | 47.7% |
| MF/Label | 3076 | 105 | 3.4% |
| SCI/Label | 0 | 0 | 0 |
| GLADNet/Label | 2956 | 112 | 3.7% |
| RetinexNet/Label | 71 | 14 | 19.7% |
| LIME/Label | 3251 | 131 | 4.0% |
| Our (Supervised)/Label | 481 | 123 | 25.6% |
| Our (Unsupervised)/Label | 454 | 110 | 24.2% |

**Table 6.** Comparison of Image Noise Level of Different Algorithms.

| Image | Local Variance Noise Estimation |
|---|---|
| Input | 0.000063 |
| Label | 0.00027 |
| KinD | 0.00150 |
| Zero-DCE | 0.00007 |
| MBLLEN | 0.00174 |
| EnlightenGAN | 0.00115 |
| SIRE | 0.00116 |
| MF | 0.01001 |
| SCI | 0.00003 |
| GLADNet | 0.00926 |
| RetinexNet | 0.01024 |
| LIME | 0.01711 |
| Our (Supervised) | 0.00389 |
| Our (Unsupervised) | 0.00422 |

From the first row of Table 7, it can be seen that the medium-light images and reference images themselves had a relatively high matching rate, which we used as the standard for evaluating matching rates. In comparison, it can be found that the matching rates of KinD, Zero-DCE, EnlightenGAN, SIRE, MF, SCI, RetinexNet, and LIME were all lower than that of the images before enhancement. We believe that this may be because these methods have destroyed the original features of the images or caused overexposure problems during the enhancement process, such as SIRE's matching rate being only 10.1%, which is significantly lower than the evaluation standard. As shown in Figure 13, the image enhanced by SIRE did indeed have a significant overexposure problem. Among all the methods, the MBLLEN method had the best effect on improving the matching rate, and our method (supervised) was second only to MBLLEN, achieving good improvement results. Therefore, it can be seen that our proposed method effectively alleviates the problem of image over-enhancement.

**Table 7.** Comparison of image matching results enhanced by different algorithms.

| Match Mode | Feature Points | Match Points | Match Rate |
|---|---|---|---|
| Input/Label | 468 | 229 | 48.9% |
| Label/Label | 649 | 649 | 100% |
| KinD/Label | 1174 | 379 | 32.2% |
| Zero-DCE/Label | 1012 | 386 | 38.1% |
| MBLLEN/Label | 750 | 483 | 64.5% |

**Table 7.** *Cont.*

| Match Mode | Feature Points | Match Points | Match Rate |
|---|---|---|---|
| EnlightenGAN/Label | 949 | 362 | 38.1% |
| SIRE/Label | 1036 | 105 | 10.1% |
| MF/Label | 1288 | 381 | 29.6% |
| SCI/Label | 885 | 397 | 44.9% |
| GLADNet/Label | 779 | 419 | <span style="color:red">53.7%</span> |
| RetinexNet/Label | 802 | 134 | 16.7% |
| LIME/Label | 1320 | 372 | 28.2% |
| Our (Supervised)/Label | 905 | 482 | 53.3% |
| Our (Unsupervised)/Label | 797 | 409 | 51.3% |

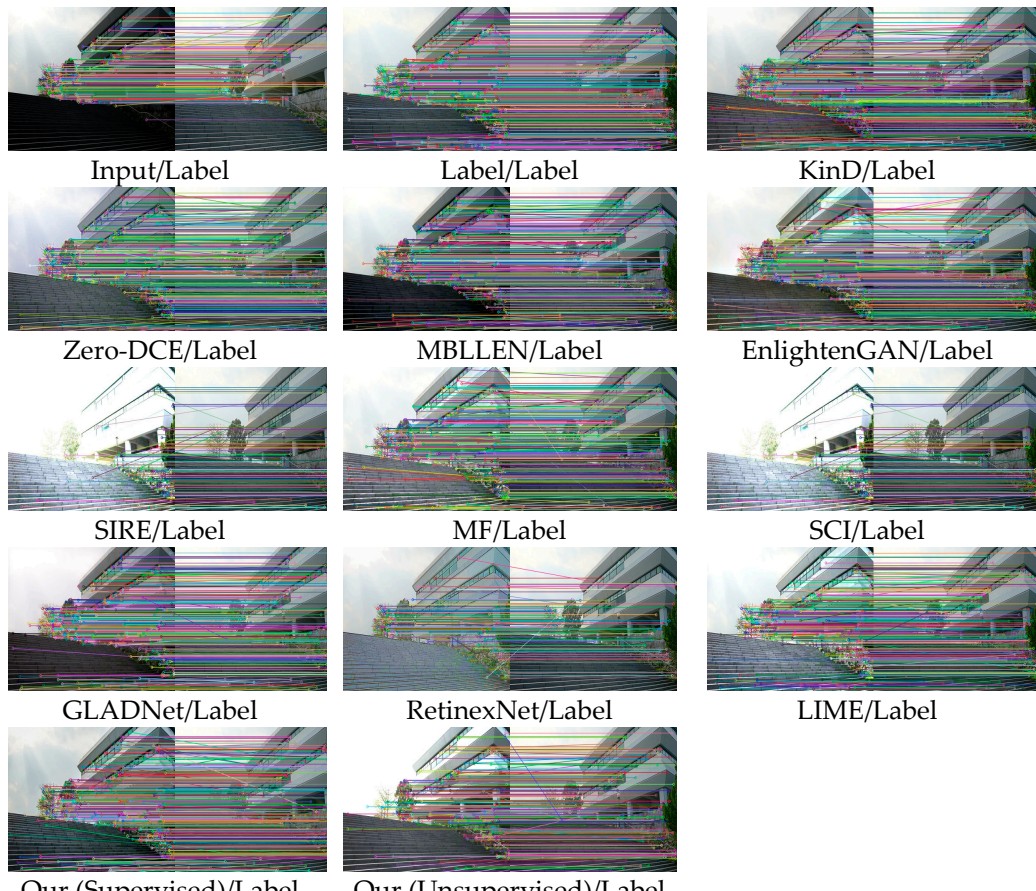

**Figure 13.** Visualization of image matching results enhanced by different algorithms.

## 5. Conclusions

In this study, we proposed a novel image enhancement network model, which is based on the LSGAN framework and can be trained in both unsupervised and supervised manners. The network model consists of a generator and a discriminator, where the generator employs a U-Net as the backbone network and introduces our proposed Attention Map Network and MBCMHSA-Net. The Attention Map Network serves as a guiding module for the backbone network and is a key network in enhancing image brightness and contrast. The MBCMHSA-Net is used to enhance image details and allocate weights for important features. Additionally, to address the problem of unstable training of traditional neural network discriminators, we proposed an improved discriminator network, which includes a convolutional encoder, a Transformer encoder, and a fully connected neural network. The experimental results show that this discriminator can better guide the training of the generator.

In the experiment section, we performed image quality evaluation experiments to test the image enhancement effects on low and extremely low illumination images. Compared to other methods, the images generated by our method had better visual effects and quality evaluation indicators. In the application experiment, our method also improved the image matching results under various lighting conditions. However, we found some limitations in our method, including:

(1) The enhancement effect of our method still needs to be improved in extremely low illumination images with very little information. These images not only have extremely low brightness and contrast but also have a lot of noise and information loss. We will continue to improve our method to make it more suitable for extremely low illumination environments.

(2) The network parameters proposed in our method are relatively large. It is necessary to optimize them further in the current trend of network lightweight.

Our future research will not only improve the limitations mentioned above but also focus on practicality to enhance the role of image enhancement methods in low-light image matching, object detection, and object tracking tasks.

**Author Contributions:** Conceptualization, T.L. and X.X. (Xucheng Xue); methodology, M.W.; software, M.W. and X.X. (Xinwei Xu); validation, M.W. and X.X. (Xinwei Xu); formal analysis, M.W. and T.L.; resources, M.W.; data curation, M.W. and X.X. (Xinwei Xu); writing—original draft preparation, M.W.; writing—review and editing, M.W. and X.X. (Xucheng Xue); project administration, T.L.; funding acquisition, T.L. All authors have read and agreed to the published version of the manuscript.

**Funding:** This work was supported by the Research Fund of the National Natural Science Foundation of China (NSFC): Project Number: 62005280, Project Leader: Taiji Lan.

**Data Availability Statement:** Publicly available datasets were analyzed in this study. This data can be found here: https://flyywh.github.io/CVPRW2019LowLight/ (accessed on 1 January 2022).

**Conflicts of Interest:** The authors declare no conflict of interest.

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
