# Peer review of "Unsupervised Image Enhancement Method Based on Attention Map Network Guidance and Attention Mechanism"

_electronics, doi:10.3390/electronics12081887_

Round 1

Reviewer 1 Report

This paper proposes to use attention map network guidance and attention mechanism for low-light image enhancement, which could be either supervised or unsupervised. The performance of the proposed scheme looks promising and the topic is of interest to the image processing community, but the paper could be further polished before being published. Here are a few suggestions:

- The English writing could be further improved. Just a few examples: in line 45, its->their/corresponding; line 7, similar->related; line 62, Method of Image Enhancement -> Image Enhancement Methods; line 167, Contribution to This Paper -> Contribution of This Paper ...

- In Section 2, the review of related works looks unbalanced. For instance, since retinex-based techniques are more recent and from current point of view more important than histogram-based methods, it makes sense to use comparable space to review this line of research and mention the latest related works (e.g., "Enhancing Low-Light Color Image via L0 Regularization and Reweighted Group Sparsity" in IEEE Access 2021). 

- Also in Section 2, it may not be necessary to separate each paragraph into a subsection. 

- In Table 1, it could be helpful to highlight the best (perhaps also the 2nd best) scores. 

Reviewer 2 Report

This research article proposed an improved method for enhancing nighttime images based on attention map network guidance and attention mechanism. To improve the quality of the paper, the authors need to consider the following comments for the revision.

Major comments

1. It is suggested that the author revises and check English writing.

2. In abstract, it is advised to add some major findings from the study and some key drawbacks/scope remain for further study.

3. The introduction is well explained with review literature. But it is not clear how and why author select two primary contributions and specify the objective based on the problem focused in literature. Is it only the image quality? Or complex environment? Or more issues (such as positioning, use of sensors, etc.)?

4. In section 3, author only describe the method. The methods are well flown. Author also explain only the datasets. But what about other type of datasets? Is the algorithm only work specific datasets?

5. Moreover, image acquisition conditions are also missing. How author define “nighttime”? Need a detailed explanation related to datasets.

6. Section 4.4 looks like the only results section. Moreover, no detailed explanation or comparison were shown in the results. It is needed to compare the results from current method to those in the literature. Also, low light and normal light are shown in the result. What about others conditions, like very low light, high light, or moderate condition, etc. These parts are missing. It would be interesting if author add and explain in details.

7. In table 2, only 5 image result were presented, and the match rate are different from each other? What is the reason behind it, although all are low-light /enhanced images?

8. Revise the conclusion explaining the methods, findings and limitation of the work with future scopes.

9. Add major findings and future trends in abstract section.

Minor comments:

1. Lines 39, 45 im-age >>image. Is it?

2. Lines 449, Tab. 2 >> table 2.

3. Lines 351-353, Abbreviation is required for all the short form used in the text for the first time. Please revise.

Round 2

Reviewer 2 Report

The author improved the manuscript well. As this is image processing work, it is always important to focus on the data collection condition and it's method.

Please carefully check the English spelling and sentence structure.

And check the author guidelines for the manuscript revision.